# Regulation of a PRMT5/NF-κB Axis by Phosphorylation of PRMT5 at Serine 15 in Colorectal Cancer

**DOI:** 10.3390/ijms21103684

**Published:** 2020-05-23

**Authors:** Antja-Voy Hartley, Benlian Wang, Guanglong Jiang, Han Wei, Mengyao Sun, Lakshmi Prabhu, Matthew Martin, Ahmad Safa, Steven Sun, Yunlong Liu, Tao Lu

**Affiliations:** 1Department of Pharmacology and Toxicology, Indiana University School of Medicine, 635 Barnhill Drive, Indianapolis, IN 46202, USA; hartleya@iu.edu (A.-V.H.); xiaohan1119@gmail.com (H.W.); sun19@iu.edu (M.S.); prabhul@iu.edu (L.P.); mm217@iu.edu (M.M.); asafa@iu.edu (A.S.); wzstevensun@gmail.com (S.S.); 2Center for Proteomics and Bioinformatics, Case Western Reserve University, Cleveland, OH 44106, USA; Benlian.Wang@case.edu; 3Department of Medical & Molecular Genetics, Indiana University School of Medicine, Indianapolis, IN 46202, USA; ggjiang@iu.edu (G.J.); yunliu@iu.edu (Y.L.); 4Department of Biochemistry and Molecular Biology, Indiana University School of Medicine, 635 Barnhill Drive, Indianapolis, IN 46202, USA

**Keywords:** colorectal cancer, NF-κB, phosphorylation, PRMT5, serine

## Abstract

The overexpression of PRMT5 is highly correlated to poor clinical outcomes for colorectal cancer (CRC) patients. Importantly, our previous work demonstrated that PRMT5 overexpression could substantially augment activation of the nuclear factor kappa B (NF-κB) via methylation of arginine 30 (R30) on its p65 subunit, while knockdown of PRMT5 showed the opposite effect. However, the precise mechanisms governing this PRMT5/NF-κB axis are still largely unknown. Here, we report a novel finding that PRMT5 is phosphorylated on serine 15 (S15) in response to interleukin-1β (IL-1β) stimulation. Interestingly, we identified for the first time that the oncogenic kinase, PKCι could catalyze this phosphorylation event. Overexpression of the serine-to-alanine mutant of PRMT5 (S15A), in either HEK293 cells or CRC cells HT29, DLD1, and HCT116 attenuated NF-κB transactivation compared to WT-PRMT5, confirming that S15 phosphorylation is critical for the activation of NF-κB by PRMT5. Furthermore, the S15A mutant when compared to WT-PRMT5, could downregulate a subset of IL-1β-inducible NF-κB-target genes which correlated with attenuated promoter occupancy of p65 at its target genes. Additionally, the S15A mutant reduced IL-1β-induced methyltransferase activity of PRMT5 and disrupted the interaction of PRMT5 with p65. Furthermore, our data indicate that blockade of PKCι-regulated PRMT5-mediated activation of NF-κB was likely through phosphorylation of PRMT5 at S15. Finally, inhibition of PKCι or overexpression of the S15A mutant attenuated the growth, migratory, and colony-forming abilities of CRC cells compared to the WT-PRMT5. Collectively, we have identified a novel PKCι/PRMT5/NF-κB signaling axis, suggesting that pharmacological disruption of this pivotal axis could serve as the basis for new anti-cancer therapeutics.

## 1. Introduction

Colorectal cancer (CRC) remains one of the leading causes of cancer-related deaths worldwide. Unfortunately, the incidence and mortality rate of this cancer is expected to rise even further, creating an urgent need to identify specific molecular pathways underlying CRC malignancy that can be targeted for the development of new therapeutic approaches. Recently, protein arginine methyltransferase 5 (PRMT5) has emerged as a key positive modulator of CRC malignancy and its overexpression is highly correlated to poor patient prognosis [1]. As a member of the superfamily of PRMTs, PRMT5 symmetrically dimethylates the guanidino nitrogen atoms of arginine residues using S-adenosylmethionine (AdoMet) as a methyl donor. Our group and others have previously reported that overexpression of PRMT5 enhanced the proliferative, migratory, and colony-forming abilities of CRC cells, whereas knockdown or pharmacological inhibition of PRMT5 had the opposite effect, making it a highly attractive therapeutic target [1,2]. Moreover, accumulating evidence suggests that PRMT5 exerts its pro-tumor functions through methylation of several pleotropic transcription factors (TFs). Through modulation of these TFs, PRMT5 can enact a wider repertoire of transcriptional changes related to many essential cellular processes that go awry during cancer progression [2,3,4,5]. In this regard, we provided the first evidence of PRMT5 dimethylating the p65 subunit of nuclear kappa B (NF-κB) at arginine 30 (R30) in response to interleukin-1β (IL-1β). This modification potently activated NF-κB signaling, resulting in enhanced DNA-binding ability of p65 and induction of more than 75% of NF-κB target genes involved in inflammation, cell proliferation, survival, and differentiation [6].

Aberrant activation of the NF-ĸB signaling pathway remains one of the most prominent etiological factors in CRC malignancy [7,8]. As a master regulator of innate and inflammatory responses, many key cellular processes that become deregulated in cancer are critically influenced by NF-ĸB-dependent transcription of a diverse array of target genes [9,10]. The family of NF-ĸB proteins is divided into two subgroups, comprising of five members. The Rel proteins, which include RelA (p65), RelB and cRel, harboring transactivation domains in their C-termini while the NF-κB1 (p50 and precursor p105) and NF-κB2 (p52 and precursor p100) proteins contain C-terminal ankyrin repeats. These proteins typically exist as dimers (p65:p50 or RelB:p52). In the absence of an activating stimulus, NF-κB is sequestered in the cytoplasm through their complex with the inhibitor of NF-κB (IκB). Activation of the classical NF-κB signaling pathway which involves the prototypical p65:p50 heterodimer, is initiated by a range of extracellular signals including pro-inflammatory cytokines such as IL-1β. These initiating signals then trigger a cascade of events that culminate in phosphorylation of IκBα by the IκB kinase (IKKα/IKKβ) complex, leading to proteasomal ubiquitin-dependent IκBα degradation. Finally, the subsequent liberation of the p65:p50 heterodimer results in its nuclear translocation and activation of NF-κB target gene transcription [10].

Although we have previously shown that PRMT5 can dynamically regulate p65 via IL-1β-inducible methylation at R30, the precise mechanisms governing this novel PRMT5/p65 signaling axis are still not fully understood. In the present study, we have now uncovered a novel mechanism that PRMT5-mediated activation of NF-κB is dependent on signal-induced phosphorylation of PRMT5 at serine 15 (S15) and propose that blocking this posttranslational modification (PTM) could serve the dual purpose of impairing the tumor-associated functions of PRMT5 via attenuating its activity towards NF-κB as a potential treatment strategy for CRC [11,12]. Compared to WT-PRMT5, substitution of serine with alanine at S15 (S15A) dramatically reduced NF-κB transactivation potential which correlated with decreased target gene transcription through attenuated recruitment of p65 to the proximal promoter of its target genes. Moreover, by demonstrating that the S15A mutant of PRMT5 disrupted its interaction with p65, our work provides new insight into the regulation of the PRMT5/NF-κB axis through phospho-dependent protein–protein interactions. These results suggest that crosstalk between kinases and PRMT5 may play a pivotal role in modulating the diverse cellular functions of PRMT5, implicating the potential utility of relevant kinase inhibitors to disrupt PRMT5-NF-κB cooperativity and impede cancer cell growth.

## 2. Results

### 2.1. Identification of Serine 15 (S15) as A Novel Phosphorylation Site on PRMT5

PTMs such as phosphorylation remain some of the key mechanisms employed by cells to not only increase the functional diversity of the proteome but to ultimately influence various aspects of pathogenesis [13]. Using mass spectrometry approaches, we screened for PTMs on PRMT5. As shown in Figure 1A, a mass shift of 80 Da was identified on S15 of the FLAG-tagged PRMT5 protein purified by FLAG-based immunoprecipitation from IL-1β-treated HEK293 cells, corresponding to the addition of a phosphorus group on this residue. To elucidate the biological role of S15 phosphorylation, we successfully overexpressed the S15A mutant at a level comparable with WT-PRMT5 in HEK293 cells and a panel of CRC cell lines, including HT29, DLD1, and HCT116 cells (Figure 1B). Further confirmation of whether S15 phosphorylation constitutes a major serine phosphorylation site on PRMT5 revealed that upon treatment of HEK293 and HT29 with IL-1β, the immunoprecipitated FLAG-WT-PRMT5 was markedly phosphorylated, whereas the PRMT5 FLAG-S15A mutant exhibited nearly completely abolished serine phosphorylation (Figure 1C). Furthermore, alignment of PRMT5 sequences showed that this S15 site is well conserved across different species (Figure 1D).

### 2.2. Phosphorylation of PRMT5 at S15 is Critical for NF-ĸB Activation and Differentially Regulates a Subset of NF-κB Target Genes

Previously, we discovered that overexpression of PRMT5 dramatically enhanced NF-ĸB activation while knockdown showed the opposite effect [2]. Moreover, based on our finding in the current work that phosphorylation of PRMT5 at S15 could be induced by IL-1β, we speculated that this PTM may also be important to the activation of NF-κB by PRMT5. Hence, using our established cell lines overexpressing WT-PRMT5 or the S15A mutant, we assessed the activation of NF-κB using a luciferase reporter assay in HEK293, HT29, DLD1, and HCT116 cells. As shown in Figure 2A, overexpression of WT-PRMT5 (WT) augmented NF-κB activity compared to the vector control (Ctrl), whereas the S15A mutant cells exhibited significantly reduced NF-κB activation compared to WT in IL-1β-inducible manner. Moreover, we further determined whether NF-κB-dependent target gene expression could be upregulated by WT-PRMT5 overexpression and compromised by the S15A mutant. HEK293 cells with or without (Ctrl) the overexpression of WT-PRMT5 or S15A, in the presence or absence of IL-1β treatment, were used to carry out Illumina microarray analyses. We observed that of the pool of IL-1β-inducible genes, approximately 48% were further upregulated by 1.5-fold or more compared to the vector control group (WT + IL-1β/Ctrl + IL-1β ≥ 1.5 fold) (Figure 2B, left pie chart). However, compared with WT-PRMT5 cells, approximately 39% of IL-1β-induced NF-κB target genes showed a two-fold or more reduction in expression in S15A cells (S15A + IL-1β/WT + IL-1β ≤ 0.5 fold) (Figure 2B, right pie chart). Furthermore, we uncovered that these genes include a range of cytokines (e.g., tumor necrosis α (TNFα), interleukin 8 (IL8)), chemokines (e.g., CCL20, CXCL10), and cell adhesion molecules (e.g., Selectin. VCAM-1), all components implicated in cancer progression (Figure 2B, bottom panel). A full list of these genes is shown in S3. Confirmation of the array data by qRT-PCR analysis revealed that the mRNA transcript levels of CCL20 and IL8 in HEK293 and a panel of CRC cells were further augmented by the overexpression of WT-PRMT5 under IL-1β-stimulating conditions whereas these transcripts were significantly attenuated by S15A overexpression (Figure 2C). Fascinatingly, in a cross comparison, we observed that several NF-κB target genes including IL8 and CCL20 were among those downregulated by the previously reported R30A mutation of p65 [6], suggesting a potential correlation between S15 phosphorylation of PRMT5 and R30 methylation of p65 dependent gene regulation.

Next, we sought to identify the signature gene networks associated with the subset of genes differentially regulated by S15A using Ingenuity Pathway Analysis (IPA). Interestingly, we observed an enrichment of terms associated with key “Biological Functions” such as “migration of tumor cells”, “proliferation of tumor cells”, and “colony formation” (Figure 2D, left panel). Moreover, networks related to “cancer” and “development disorders” were among the top enriched “Disease” networks (Figure 2D, right top panel) while IL-1β, the NF-ĸB complex and IKBKB were among the most highly enriched “Upstream Regulators” (Figure 2D, right bottom panel). Furthermore, representative networks revealed NF-κB as a key interaction node among the genes upregulated by WT-PRMT5 and compromised by the S15A mutant (Figure 2E and Appendix A). Collectively, these data support the notion that phosphorylation of PRMT5 at S15 can augment NF-ĸB signaling via regulation of p65 transactivation potential and a subset of NF-κB target genes whose functions are pro-inflammatory and cancer-related in nature.

### 2.3. S15A Shows Similar Subcellular Compartmentalization to WT-PRMT5 and Attenuates NF-ĸB Activation Independent of IĸBA Degradation and p65 Nuclear Translocation

We also wondered whether the reduced transcriptional activation of NF-ĸB by S15A may be due to other factors such as altered subcellular localization of S15A compared to WT-PRMT5, modulation of the degradation pattern of IĸBα, or blockade of the nuclear translocation of p65. Our data showed no observable difference in the subcellular distribution pattern of the WT-PRMT5 and S15A proteins (Appendix A). Moreover, no difference in the translocation of p65 to the nucleus (Appendix A) or IĸBα degradation pattern was detected between WT-PRMT5 and S15A overexpressing HEK293 cells (Appendix A), indicating that S15A may be acting independently of these mechanisms.

### 2.4. The S15A Mutant Interacts Less Well with p65 and Attenuates Promoter Occupancy of p65 at NF-ĸB Target Genes

To further explore the possible mechanisms underlying the downregulation of NF-ĸB activity by S15A, we wondered whether this mutant altered interaction between PRMT5 and p65 and thus would impair the downstream transcriptional competence of p65. As shown in Figure 3A and consistent with our previous findings [6], co-immunoprecipitation of FLAG-WT-PRMT5 (Flag-WT) and p65 showed enhanced interaction of PRMT5 with p65 in IL-1β treated HEK293 and HT29 cells. On the other hand, Flag-S15A substantially attenuated this interaction, providing the first evidence that this IL-1β-inducible PRMT5/p65 interaction node occurs in a phospho-dependent manner (Figure 3A).

To further elucidate the mechanistic underpinnings for the observed S15A-mediated downregulation of NF-ĸB target genes, we conducted chromatin immunoprecipitation (ChIP)-PCR analysis to determine the proximal promoter occupancy of p65 at the IL8 gene, a prototypical target gene of NF-ĸB that harbors a ĸB element in its promoter (Figure 3B). Excitingly, when compared to the Ctrl cells, overexpression of WT-PRMT5 led to a strikingly enhanced occupancy of p65 at the IL8 promoter along a time course treatment with IL-1β in both HEK29 and HT29 cells, while overexpression of S15A remarkably reduced this effect (Figure 3C).

### 2.5. Phosphorylation of PRMT5 at S15 Controls the Proliferative, Migratory, and Anchorage-Independent Growth Abilities of CRC Cells

The aforementioned microarray and IPA data suggested that S15A-downregulated genes were involved in cancer-associated biological functions (Figure 2D). Moreover, we previously demonstrated that overexpression or knockdown of PRMT5 could promote or impede, respectively, CRC cell growth, migration, and anchorage-independent growth. Collectively, these findings prompted us to further probe the functional significance of overexpressing the S15A mutant on the same cancer properties. First, we demonstrated that compared to Ctrl group, overexpression of WT-PRMT5 in CRC cells could significantly enhance cell growth whereas overexpression of S15A attenuated this effect compared to WT (Figure 4A). Next, we tested whether the S15A mutant could affect the anchorage-independent growth ability in CRC cells. As shown in Figure 4B, compared to Ctrl group, WT-PRMT5-overexpressing cells showed a significant increase in both the size and number of colonies formed whereas the S15A-overexpressing cells formed much less and significantly smaller colonies (Figure 4B). Finally, we examined the role of S15 phosphorylation in regulating CRC cell migratory capacity. Unsurprisingly, when compared to Ctrl group, overexpression of WT-PRMT5, consistent with previous findings, resulted in a significant increase in the migrated cells whereas overexpression of the S15A mutant had the opposite effect (Figure 4C). Taken together, these data strongly suggest that the enhanced proliferative, migratory, and anchorage-independent growth abilities of CRC cells mediated by PRMT5, is at least in part, facilitated by S15 phosphorylation.

### 2.6. S15 Phosphorylation Regulates IL-1β-Inducible PRMT5 Methyltransferase Activity

Based on the domain architecture of human PRMT5, we determined that the S15 residue is located within the unique N-terminal triose phosphate isomerase (TIM) barrel (Figure 5A). Interestingly, other studies suggest that contributions from the TIM-Barrel may potentially control PRMT5 oligomerization, substrate specificity, and enzymatic activity. We therefore speculated that phosphorylation of PRMT5 at S15 could regulate its methyltransferase activity. To test this hypothesis, we immunoprecipitated enzyme preps of FLAG-WT-PRMT5, FLAG-S15A, and FLAG-E444D (Glu 444 to Asp mutant)—an enzymatically dead mutant from HEK293 cells, that were treated or left untreated with IL-1β. We then employed our previously described AlphaLISA assay that tests the specific activity of PRMT5 using a histone H4R3 peptide as a substrate. As shown in Figure 5B, IL-1β treatment enhanced the activity of WT-PRMT5 by approximately two-fold while S15A showed significant attenuation of this activity. Furthermore, there was no significant difference between the activity of the WT and S15A enzymes under basal conditions, suggesting a putative IL-1β-dependent structural mechanism by which S15 phosphorylation regulates PRMT5′s methyltransferase activity albeit other factors are likely at play.

### 2.7. Knockdown of PKCι Attenuates Phosphorylation of PRMT5 and Disrupts PRMT5-Mediated NF-ĸB Signaling

To determine which kinase(s) phosphorylates PRMT5 at S15, we utilized the Human Protein Reference Database (http://www.hprd.org/ PhosphoMotif finder) to predict PKC and PKA as potential kinases related to S15 phosphorylation based on site-specific consensus phospho-motifs (Figure 6A). Next, to narrow down which kinase was the most promising candidate, we employed mass spectrometry identification studies to determine top kinases that were immunoprecipitated with WT-PRMT5. Among these, we identified PKCι as a PRMT5-interacting partner, which was further verified by co-immunoprecipitation in HEK293 and HT29 cells (Figure 6B). We further determined whether modulating PKCι expression could affect serine phosphorylation levels on PRMT5 by using a pool of shRNA constructs to stably knockdown the expression of PKCι with or without stable co-expression of WT-PRMT5 (Flag-WT) in HEK293 cells (Figure 6C). These cells were then treated with IL-1β followed by immunoprecipitation of Flag-WT. As shown in Figure 6D, a significant induction of serine phosphorylation of PRMT5 was observed under IL-1β-stimulating conditions in the cells dually expressing Flag-WT and shscramble constructs. In contrast, knockdown of PKCι cells treated with IL-1β correlated with diminished phosphorylation of Flag-WT, implicating PKCι as a promising candidate kinase.

Our next logical step was to examine whether PKCι could in turn regulate NF-ĸB signaling through disruption of the PRMT5/p65 interaction and/or PRMT5-mediated transactivation of NF-ĸB. Stable HEK293 cells with expression of vector control, WT-PRMT5, or S15A constructs with or without shscramble or shPKCι were established (Figure 6C) and subsequently subjected to co-immunoprecipitation (Figure 6E) and NF-ĸB luciferase assays (Figure 6F). As shown in Figure 6E, knockdown of PKCι correlated with a disruption of the IL-1β-inducible PRMT5/p65 interaction, similar to that previously observed with the S15A mutant (Figure 3A). Furthermore, luciferase assays revealed that upon knockdown of PKCι, NF-κB activity in the vector control and WT-PRMT5-overexpressing cells was significantly attenuated compared to the respective shscramble counterparts. Interestingly, no significant further decrease in NF-κB activity in the IL-1β-treated S15A/shPKCι cells was observed, suggesting that blockade of PKCι-regulated PRMT5-mediated activation of NF-κB was likely through phosphorylation of PRMT5 at S15 (Figure 6F). A similar phenomenon was observed when we used a selective small molecule PKCι inhibitor, CRT0066854, which disrupted IL-1β-inducible WT-PRMT5-mediated activation of NF-ĸB luciferase activity whereas no notable further decrease in NF-ĸB activity was detected in S15A cells treated with the inhibitor (Figure 6G). Collectively, these data suggest that PKCι may act upstream of PRMT5 and NF-ĸB. Interestingly, as shown in Figure 6H, transcript levels of PKCι across colorectal adenocarcinoma (COAD) tumors are significantly upregulated in human CRC samples compared to normal tissue, suggesting that PKCι could be a potential biomarker for CRC diagnosis.

### 2.8. Knockdown or Small-Molecule Inhibition of PKCι Attenuates NF-ĸB Activation, NF-ĸB Target Gene Expression, Cell Growth, Anchorage-Independent Growth, and Cell Migration

To further explore the functional significance of PKCι blockade in our CRC cells, we generated HT29, DLD1, and HCT116 CRC cells expressing either shscramble or shPKCι constructs (Figure 7A). Next, we used these cells to determine NF-ĸB transactivation and target gene expression in the absence and presence of IL-1β stimulation. As shown in Figure 7B, luciferase assays revealed that upon knockdown of PKCι, NF-κB activity in the shPKCι cells was significantly attenuated compared to their shscramble counterparts. Furthermore, selective small molecule inhibition of PKCι with CRT0066854 significantly disrupted IL-1β-inducible activation of NF-ĸB luciferase activity with increasing µM concentrations of inhibitor compared to the vehicle control CRC cells (Figure 7C). Next, we assessed the relative mRNA levels of NF-ĸB target genes CCL20 and IL8 using qPCR analyses and observed that compared to shscramble cells, knockdown of PKCι correlated with significant decreases in IL-1β-inducible expression of these genes (Figure 7D). Moreover, a similar phenomenon was observed in CRC cells treated with PKCι inhibitor CRT0066854 (Figure 7E).

Previously, we showed that PRMT5 overexpression augmented certain cancer-associated biological functions in CRC cells in an S15 phosphorylation-dependent manner (Figure 4). Upon identifying PKCι as a mediator of this phosphorylation event on PRMT5, we logically sought to determine whether depletion or inhibition of PKCι would also correspondingly decrease the growth, colony formation, and migration of CRC cells. As shown in Figure 8, compared to shscramble groups, shRNA-mediated depletion of PKCι resulted in a significant decrease in the cell number (Figure 8A, top panel). Similarly, treatment with PKCι inhibitor CRT0066854 decreased the overall growth of CRC cells over 7 days in culture (Figure 8A, bottom panel). Next, we assessed the effect of PKCι knockdown or inhibition on CRC anchorage-independent growth and migration. Figure 8B, top panel shows that colonies formed in shPKCι-expressing CRC cells were significantly smaller than shscramble counterparts. Likewise, treatment with PKCι inhibitor CRT0066854 showed a similar phenomenon (Figure 8B, bottom panel). Finally, we examined the effect of PKCι knockdown on CRC cell migratory capacity. Consistent with previous findings, a significant decrease in the number of migrated cells was observed in shPKCι cells when compared to shscramble CRC cells (Figure 8C). Taken together, these data strongly suggest that PKCι positively regulates NF-ĸB activation, target gene expression and in part, regulates the proliferative, migratory, and anchorage-independent growth abilities of CRC cells.

### 2.9. Hypothetical Model

Based on the findings from this study, we provide evidence that IL-1β stimulation induces PKCι-mediated phosphorylation of PRMT5 at S15, an important step in the transactivation of NF-κB and ultimately activation of target gene expression. Mechanistically, phosphorylation of PRMT5 at S15 mediates the PRMT5-p65 interaction and proximal promoter occupancy of p65 at its target genes (e.g., IL8) and thus constitute pivotal mechanisms by which PRMT5 can fine-tune NF-κB activation and target gene expression. Furthermore, S15 phosphorylation mediates IL-1β-induced PRMT5 activity. Together, these potentially serve to facilitate the tumor-associated functions exerted by this PKCι-mediated PRMT5-dependent NF-κB activation, including the enhanced proliferation, anchorage-independent growth, and migration associated with PRMT5 overexpression (Figure 9).

## 3. Discussion

A wealth of reported findings supports the essential biological functions of PRMT5 in cell proliferation, transcriptional activation, signal transduction, and cell differentiation through methylation of its substrate proteins [14]. However, far less is known about the factors that underlie its involvement in various signaling axes or the mechanisms that fine-tune its activity towards certain substrates. Nevertheless, this knowledge is critical since these mechanisms form the basis for devising novel therapeutic strategies for targeting PRMT5. In the present study, we report a novel phosphorylation modification on PRMT5 at S15 and mechanistically validated its role in positively regulating the PRMT5/NF-κB signaling node through modulation of the PRMT5/p65 interaction, NF-κB target gene expression and proximal promoter occupancy of p65 at its target genes as well as the IL-1β-inducible methyltransferase activity of PRMT5. Moreover, our data indicate that the S15A mutant of PRMT5 could impede the proliferative, migratory, and anchorage-independent growth capabilities of CRC cells. This evidence supports a model in which phosphorylation of PRMT5 may act as a pivotal step in enhancing the NF-κB response observed with PRMT5 overexpression which involves upregulation of the transcription of cancer-associated NF-κB target genes.

Phosphorylation of PRMT5 on other types of amino acids, such as tyrosine and threonine was previously reported. For instance, the Liu group identified tyrosine phosphorylation (Y297, 304, 307) of PRMT5 by mutant Jak2 playing an inhibitory role on PRMT5′s dimethylation of histone substrates. This inhibition also correlated with an enhanced myeloproliferative phenotype [15]. In a quite different context, threonine phosphorylation of PRMT5 was shown to modulate PRMT5′s function by triggering a PDZ/14-3-3 interaction switch [16]. In contrast, in our case, we showed for the first time that PRMT5 can be phosphorylated on a serine residue upon IL-1β treatment, leading to the enhancement of its methyltransferase activity and interaction between PRMT5 and its substrate, p65. This sheds important light on how the same but context-dependent modification on different amino acid residues can produce varied outcomes. Intriguingly, our data also implies that cytokines in the tumor microenvironment may affect cancer progression since IL-1β is a pleiotropic pro-inflammatory cytokine abundantly secreted within the tumor microenvironment and critically mediates the induction of a local network of cytokines/chemokines that modulate tumor growth and invasiveness [17].

Furthermore, the discovery of PKCι’s link to phosphorylation of PRMT5 on S15 and NF-κB activation is significant. PKCι is an oncogene that has been implicated in the initiation and progression of CRC via regulation of epithelial cell integrity and polarity in a model of colitis-associated colon cancer [18]. Moreover, mutations and genetic alterations of PKCι are often detected across various cancer types, making it a highly attractive therapeutic target [19,20]. Interestingly, we showed that PKCι is also upregulated in CRC patient tissues (Figure 6H) and its selective inhibition attenuated PRMT5-mediated activation of NF-κB, target gene expression, as well as CRC cell growth and migration. It is worth noting that luciferase and functional assays carried out using the CRT0066854 PKCι inhibitor (Figure 7 and Figure 8) sometimes showed a stronger effect compared to shPKCι counterparts. This effect may be due in part to the fact that CRT0066854 inhibits both PKCι and PKCζ isoforms. Moreover, the concentrations (low or high) of CRT0066854 we used may have played a role in the effect that it generated. Additionally, we may appreciate that shRNA knockdown approaches rarely generate a 100% knockdown effect. Taken together, all these potential factors may contribute to the slight differences between using these two different experimental approaches. Although we utilized both pharmacologic and shRNA knockdown approaches, we acknowledge that each approach possesses independent value toward approaching the same biological question but results may not necessarily be perfectly aligned albeit a similar appreciable trend is clear. Overall, in the future, it would be very interesting to determine the anti-tumor efficacy of inhibiting PKCι using relevant CRC mouse models and determine whether these applications could potentially be generalized to other tumor types.

From a structural perspective, PRMT5 is commonly found complexed with other proteins and these interactions have been shown to dictate both its catalytic activity and substrate specificity [21,22]. Interestingly, the N-terminal domain (residues 13–292) of PRMT5 adopts a TIM barrel structure, which has been implicated as essential for co-factor binding (e.g., MEP50) as well as acting as a scaffold for the binding of adaptor proteins (e.g., pICLn, Riok1, etc.) [21,22,23]. However, the exact mechanisms underlying structural changes in the PRMT5 protein that lead to altered activity, cofactor binding, and/or substrate selectivity are still largely unknown. In keeping with these observations, the location of the highly conserved S15 residue within the TIM Barrel region as well as its role in mediating a phospho-dependent interaction axis between PRMT5 and its substrate p65 support a possible regulatory role for this segment. Future molecular dynamics simulation as well as co-crystallization studies may provide critical structural insights into S15 phosphorylation-induced conformational changes of PRMT5. These may serve as a basis for understanding mechanisms that dictate PRMT5′s activity and discrete binding to p65 and other binding partners.

In conclusion, both NF-κB and PRMT5 are individually known to play critical roles in cancer progression, and the mechanisms underlying the cooperativity between the two are inherently critical and complicated. Based on our results, we presented a hypothetical model in which IL-1β stimulation activates the NF-κB pathway and induces PKCι-mediated phosphorylation of PRMT5. Insights from this study are, therefore, of particular importance in terms of enhancing our understanding of the role of phosphorylation sites such as S15 on PRMT5 in modulating the interaction between PRMT5 and NF-κB. In the future, these findings may also open up novel therapeutic avenues for CRC patients based on disrupting the PKCι/PRMT5/NF-κB signaling axis.

## 4. Experimental Procedures

### 4.1. Liquid Chromatography-Tandem Mass Spectrometric Analysis

Coomassie-stained SDS-PAGE gel band containing Flag-PRMT5 protein was subjected to in-gel tryptic digestion. Flag-PRMT5 gel pieces were subjected to destaining and reduction of cysteine residues using 50% acetonitrile in 100 mM ammonium bicarbonate and 100% acetonitrile followed by treatment, followed by 20 mM DTT at room temperature for 60 min. Alkylation with 55 mM iodoacetamide for 30 min was performed in the dark. The solution was removed, and the gel pieces were washed with 100 mM ammonium bicarbonate and dehydrated in acetonitrile. Gel pieces were then dried in a SpeedVac centrifuge, and proteolytically digested by rehydration overnight at 37 °C in 50 mM ammonium bicarbonate containing sequencing grade modified trypsin (Promega, Madison, WI, USA). Extracted peptides were treated with 50% acetonitrile in 5% formic acid, dried, and reconstituted in 0.1% formic acid for mass spectrometry analysis. Analysis of proteolytic digests was performed by using an LTQ Orbitrap XL linear ion-trap mass spectrometer (Thermo Fisher Scientific, Waltham, MA, USA), coupled with an Ultimate 3000 HPLC system (Dionex, Sunnyvale, CA, USA). The digests were injected onto a reverse-phase C18 column (0.075 × 150 mm, Dionex) equilibrated with 0.1% formic acid/4% acetonitrile (*v*/*v*). A linear gradient of acetonitrile from 4% to 40% in water in the presence of 0.1% formic acid over a period of 45 min was used at a flow rate of 300 nL/min. The spectra were acquired by data dependent methods, consisting of a full scan (*m*/*z* 400–2000) and then tandems on the five most abundant precursor ions. The previously selected precursor ions were scanned once during 30 s and then were excluded for 30 s. The obtained data were analyzed by Mascot software (Matrix Science, Chicago, IL, USA) against customized PRMT5 protein database with the setting of 10 ppm for precursor ions and 0.8 Da for product ions. Carbamidomethylation of cysteine was set as fixed modification, while oxidation of methionine, phosphorylation of serine, threonine, and tyrosine were set as variable modifications. The tandem mass spectra of candidate-modified peptides were further interpreted manually.

### 4.2. Cell Lines and Transfections

The HEK293C6 (HEK293) cell line was previously described [24]. CRC cell lines HT29, DLD1, and HCT116 were procured from ATCC (Manassas, VA, USA) and were cultivated in RPMI (Roswell Park Memorial Institute, Buffalo, NY, USA) 1640 medium supplemented with 100 units/mL penicillin, 100 g/mL streptomycin, and 5% fetal bovine serum (FBS). The FLAG-S15A and FLAG-E444D mutants of PRMT5 were generated using the QuikChange II XL Site-Directed Mutagenesis Kit from Agilent Technologies. Primers were designed using the Agilent Technologies QuikChange Primer Design online software. To generate stable cell lines using lentiviruses, various PRMT5 constructs including FLAG-WT-PRMT5 were cloned into the lentiviral backbone pLVX-IRES-puro vector and transfected into HEK293T packaging cell line along with relevant lentivirus packaging and helper plasmids using Lipofectamine and PLUS reagents (Life Technologies/Invitrogen, Carlsbad, CA, USA). Similarly, the pool of five shRNA constructs targeting PKCι (Sigma-Aldrich, St. Louis, MO, USA) was used to transfect HEK293T cells for lentivirus generation. HEK293C6 and CRC cell lines were subsequently infected with relevant lentiviruses as previously described [2,6]. All stable cell lines were selected with 1 μg/mL of puromycin.

### 4.3. Western Blotting and Antibodies

Cells were cultured to about 90–95% confluence before treatment with 10 ng/mL of IL-1β. Whole cell samples were collected and lysed using Radio Immunoprecipitation Assay buffer (RIPA buffer: 150 mM NaCl, 0.1% Triton X-100, 0.5% sodium deoxycholate, 0.1% sodium dodecyl sulfate (SDS), 50 mM Tris-HCl pH 8.0, and protease inhibitors). Whole cell lysates were then separated by SDS/PAGE gels, and further assessed by Western blotting. Different antibodies were used to detect the target proteins of interest, obtained from the following commercial sources: anti-PRMT5 (Abcam, Cambridge, UK; ab109451), anti-FLAG (Sigma-Aldrich, St. Louis, MO, USA; F1804), and anti-p65 (Santa Cruz Biotechnology, Santa Cruz, CA, USA; sc-109); anti-PKCι (Proteintech, Chicago, IL, USA; 66493-1-Ig). For cell fractionation experiments, cytoplasmic and nuclear fractions were subject to SDS-PAGE and probed with anti-LaminB1 (Proteintech, Chicago, IL, USA; 12987-1-AP); anti-α-tubulin (Cell Signal, Danvers, MA, USA; 2144S).

### 4.4. Luciferase Assays

NF-κB luciferase assays were conducted by transiently transfecting the κB-luciferase construct p5XIP10 κB into parental control or FLAG-PRMT5 stable cell lines after which luciferase activity was quantified 48 h later using the Luciferase Assay System with Reporter Lysis Buffer kit (Promega, Fitchburg, WI, USA; E4030). The κB-luciferase plasmid p5XIP10 κB contains five tandem copies of the NF-κB DNA binding site derived from the IP10 gene (an established target gene of NF-κB) upstream of a luciferase reporter gene. Luciferase activity was measured using a Synergy H1 Multi-Mode Reader (BioTek Instruments Inc., Winooski, VT, USA).

### 4.5. Co-Immunoprecipitations

Cells stably expressing FLAG-PRMT5 proteins were cultured to ≈95% confluency then lysed in co-immunoprecipitation buffer (1% Triton X-100 (*v*/*v*), 50 mM Tris-HCl (pH 7.4), 150 mM NaCl, 1 mM EDTA, 1 mM sodium orthovanadate, 20 µM aprotinin, 1 mM phenylmethanesulfonyl fluoride, and 1 mM pepstatin A). FLAG-PRMT5 proteins were then immunoprecipitated with anti-FLAG-M2 beads (Sigma-Aldrich, St. Louis, MO), using immunoprecipitation methods previously described [2,6]. Briefly, cell lysates with equivalent amounts of protein were incubated with anti-FLAG-M2 beads at 4 °C overnight. Beads were then washed, and FLAG-tagged proteins were eluted and separated by SDS/PAGE [2,6].

### 4.6. Cell Proliferation and Anchorage-Independent Growth Assays

For cell proliferation assays, CRC cells overexpressing FLAG-PRMT5 constructs were seeded in triplicate at 2 × 10^4^ cells/well in a 6-well plate. Cells were counted at days 3, 5, 7, and 9 post-seeding using a cell counting chamber. For anchorage-independent growth assays, type VII agarose (Sigma-Aldrich, St. Louis, MO) was used to prepare 2.4% and 1.2% bottom and top agar layers, respectively. Then, 2 × 10^5^ cells were resuspended in the top layer and plated onto the bottom layer. Cells were then cultured for 12–14 days at 37 °C and 5% CO_2_. Images of colonies were captured using a Canon EOS Rebel T3i Digital SLR camera and colony size and number were quantified using the ImageJ v.1.52a software (Madison, WI, USA; http://imagej.nih.gov/ij/).

### 4.7. Migration Assay

Migration assays were conducted using Boyden chambers. Briefly, a Boyden chamber consists of 8 μm pore size cell culture inserts in a 24-well plate. Each insert was coated with gelatin on the side facing the lower chamber. A total of 2 × 10^5^ cells were seeded in the top of the insert (upper chamber) in serum-free media while serum-rich media (10% serum) was supplied in the well below as a chemoattractant. After 48 h, migrated cells were fixed with 4% formaldehyde and stained with crystal violet. Stained cells were visualized with a light microscope and quantified. Images were captured using a Canon EOS Rebel T3i Digital SLR camera.

### 4.8. Illumina Microarrays and Quantitative PCR (qPCR)

Microarray and qPCR experiments were carried out as previously described [6]. Briefly, parental or 293 cells with WT-PRMT5 or S15A overexpression were cultured to ≈90% confluence and total RNA was isolated using Trizol reagent (Invitrogen, Carlsbad, CA, USA). Total isolated RNA was used to prepare cDNA using the SuperScript III First-Strand Synthesis PCR System (Invitrogen, Carlsbad, CA). cDNA was labeled with biotin-UTP using the Illumina Total Prep RNA amplification kit (Ambion/Applied Biosystems, Foster City, CA, USA), hybridized to Illumina Human Ref-v3 v1 Expression Bead Chips and then scanned in a Bead Array reader using standard Illumina protocols (Illumina, San Diego, CA, USA). Illumina’s Bead Studio software was used for data analysis. qPCR was carried out using FastStart Universal SYBR Green Master ROX (Roche, Basel, Switzerland). Primers were designed by the Primer Express 3.0 software. Primer information is listed below: TNF-Forward: 5′-TGGCCCAGGCAGTCAGA-3′; TNF-Reverse: 5′-GGTTTGCTACAACATGGGCTACA-3′; IL8-Forward: 5′-TCCTGATTTCTGCAGCTCTGT-3′; IL8-Reverse: 5′-AAATTTGGGGTGGAAAGGTT-3′; CCL20-Forward: 5′-GTGCTGCTACTCCACCTCTG-3′; CCL20-Reverse: 5′-CGTGTGAAGCCCACAATAAA-3′.

### 4.9. Chromatin Immunoprecipitation (ChIP)

Cells were left untreated or stimulated with 10 ng/mL of IL-1β for 1 and 4 h, and cross-linked with 1% formaldehyde for 10 min at room temperature (R.T.). The cross-linking was stopped by adding glycine and cells were then washed with cold× phosphate-buffered saline (1× PBS, 137 mM NaCl, 2.7 mM KCl, 10 mM Na2HPO4, 1.8 mM KH_2_PO_4_), scraped, and pelleted by centrifugation at 2000 rpm. Cells were lysed in Farnham Lysis buffer (mM PIPES pH 8.0/85 mM KCl/0.5% Tween 20) supplemented with protease inhibitors followed by chromatin shearing to yield fragments of 200–1000 bp using a sonifier (Fisher Scientific, Hampton, NH, USA) equipped with a microtip (40 s on/50 s off, 4 min at 40% power output). Sonicated lysates were centrifuged, and the resulting supernatant was diluted 5-fold with ChIP dilution buffer containing 16.7 mM Tris HCl pH 8.1, 167 mM NaCl, 0.01% SDS, 1.1% (*v*/*v*) Triton X-100, and 1.2 mM EDTA. Diluted lysates were precleared for 1 h with protein A/G agarose. Immunoprecipitations were performed using ChIP-grade anti-RelA (Abcam, ab7970, Cambridge, United Kingdom) antibody at 4 °C overnight. Immune complexes were collected with protein A/G agarose, washed with low-salt wash buffer [20 mM Tris HCl pH 8.1, 150 mM NaCl, 0.1% SDS, 1% (*v*/*v*) Triton X-100, and 2 mM EDTA], high-salt wash buffer [20 mM Tris HCl pH 8.1, 500 mM NaCl, 0.1% SDS, 1% (*v*/*v*) Triton X-100, and 2 mM EDTA], LiCl wash buffer [10 mM Tris HCl pH 8.1, 250 mM LiCl, 1% (*w*/*v*) sodium deoxycholate, 1% (*v*/*v*) IGEPAL-CA630, and 1 mM EDTA], and TE buffer (10 mM Tris HCl pH 8.0 and 1 mM EDTA). Protein-DNA complexes were eluted from antibodies with elution buffer containing 1% SDS and 0.1 M NaHCO3, incubated in the presence of 192 mM NaCl for 4 h at 65 °C, and digested with proteinase K for 1 h at 45 °C. DNA was recovered using the Qiagen quick DNA purification kit and IL8 gene-specific ChIP primers (Qiagen, Hilden, Germany) were used in the PCR analyses.

### 4.10. AlphaLISA Assay

FLAG-WT-PRMT5 or S15A mutant enzyme was purified from 293 cells using anti-FLAG-M2 beads (Sigma-Aldrich, St. Louis, MO) as described in co-immunoprecipitation methods above. The enzyme prep was diluted in assay buffer (30 mM Tris, pH 8.0, 1mM DTT, 0.01% BSA, 0.01% Tween-20) prior to use. SAM (New England Biolabs, Ipswich, MA, USA) and unmethylated histone H4R3 (Anaspec, Fremont, CA, USA) were used as the methyl group donor and PRMT5 enzyme substrate, respectively. Acceptor beads diluted in 1× Epigenetics buffer (PerkinElmer, Waltham, MA, USA) were added at a final concentration of 20 μg/mL to the reaction mixture and the plate was incubated at R.T. for 1 h. Donor beads diluted in 1× Epigenetics buffer (PerkinElmer, Waltham, MA) were then added at a final concentration of 20 μg/mL and the plate was further incubated at R.T. for 30 min. All reactions were performed in triplicate and repeated three independent times. The plates were read using an EnVision^®^ Reader [2].

### 4.11. Ingenuity Pathway Analysis (IPA)

Groups of genes regulated by S15A were analyzed by the IPA software 8.7 (Qiagen, Hilden, Germany). The setting and filter were as follows: reference set: Ingenuity Knowledge Base (Genes _ Endogenous Chemicals); Relationship to include: Direct and Indirect; Includes Endogenous Chemicals; Filter Summary: Consider only molecules where species _ Human OR Rat OR Mouse. The *p* values for the enrichment test were calculated using Fisher’s exact test, right-tailed. Log10 (p) was visualized to the left of the *p* value. *p* < 0.05 was considered to be statistically significant.

### 4.12. Cell Fractionation Assay

Fractionation experiments were conducted according to the manufacturer’s instructions for the nuclear extract kit (Active Motif, Carlsbad, CA, USA). Briefly, cells were grown to about 80% confluence, washed with ice-cold 1× PBS/Phosphatase inhibitors, collected and pelleted for 5 min at 500 rpm. Cell pellets were gently resuspended in 1× hypotonic buffer and incubated for 15 min on ice. The cytoplasmic fraction was collected as the lysate following centrifuging for 30 s at 14,000× *g*. The remaining cell pellet was further lysed in Complete Lysis Buffer, incubated on ice for 30 min and centrifuged for 10 min at 14,000× *g* to collect the nuclear fraction. The cytoplasmic and nuclear proteins were then subjected to Western blot analyses.

### 4.13. Statistical Analysis

Statistical analyses were performed using Prism 6 software (GraphPad, San Diego, CA, USA). Data represent the mean ±S.D. or ±SEM as indicated. A two-tailed Student’s *t*-test was used when comparing two means between groups as specified. All statistics were carried out for triplicate experiments and a *p* < 0.05 was considered statistically significant.

## Figures and Tables

**Figure 1 ijms-21-03684-f001:**
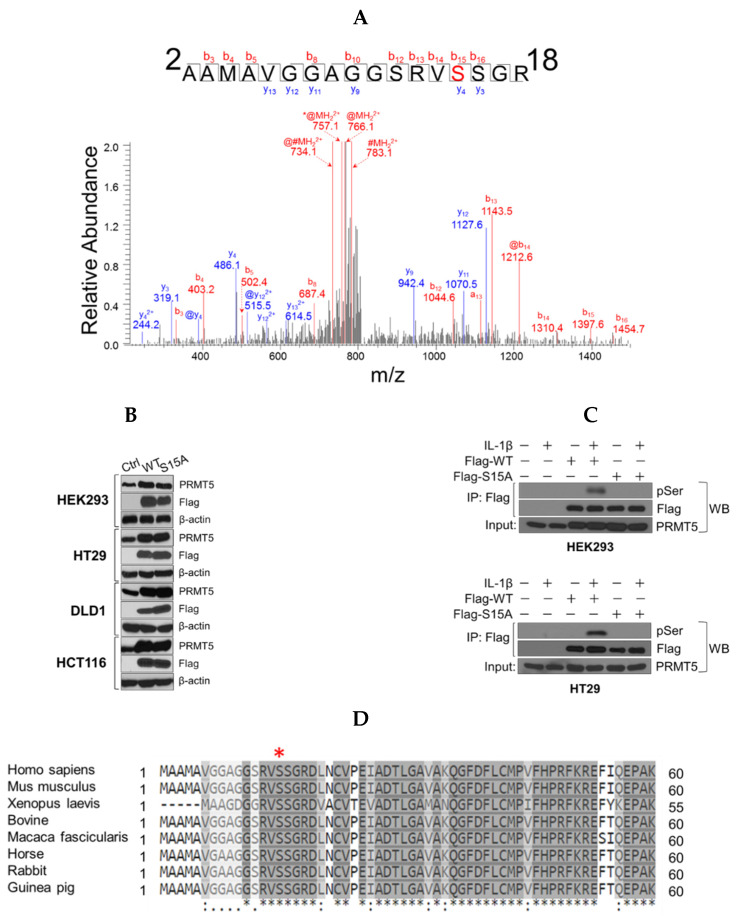
Identification of phosphorylation of Serine 15 (S15) on PRMT5. (**A**) Top panel, mass spectrometry (MS) experiment identifies S15 as a phosphorylated residue in response to IL-1β treatment. A mass shift of 80 Da was observed, indicating the existence of the phosphorylation modification. Bottom panel, Gel-code blue stained MS gel indicates a purified strong FLAG-PRMT5 protein band (left). Western blot analysis confirmation of the identity of the purified band as PRMT5 (right). (**B**) Establishment of wild type (WT) or Serine 15 to Alanine (S15A) mutant FLAG-PRMT5 overexpressing stable cells. Western blot images, showing overexpression of FLAG-PRMT5 constructs probed with anti-PRMT5, or FLAG, or β-actin respectively, in HEK293 cells or HT29, DLD1, and HCT116 colon cancer cells. (**C**) Confirmation of phosphorylation of PRMT5 at S15 using co-immunoprecipitation and Western blot analysis. Either HEK293 (top panel) or HT29 cells (bottom panel) were treated with 10 ng/mL of IL-1β or left untreated for 1 h. Samples were collected and the WT-PRMT5 (Flag-WT) or S15A (Flag-S15A) protein was further immunoprecipitated with anti-FLAG beads and subjected to Western analysis using an anti-phospho-serine motif antibody (pSER). The inputs were probed with anti-PRMT5 antibody. (**D**) Cross-species alignment of amino acid sequences from PRMT5 proteins (residues 1–60). The conserved S15 residue is indicated on the top by the red asterisk (www.uniprot.com).

**Figure 2 ijms-21-03684-f002:**
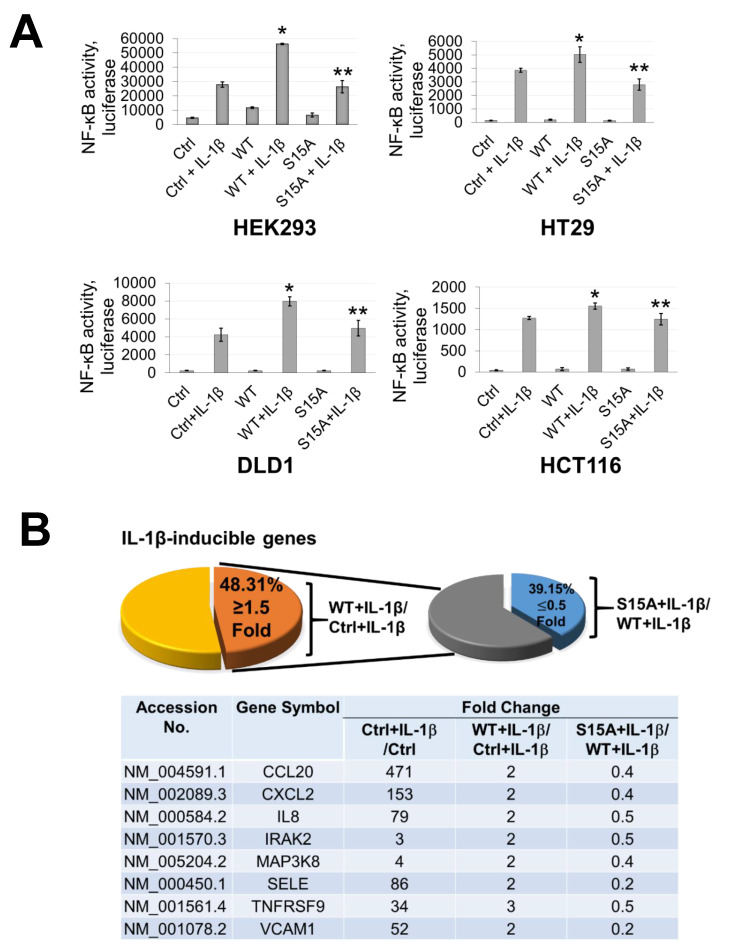
Phosphorylation of PRMT5 at S15 is critical for NF-ĸB activation and differentially regulates a subset of NF-κB target genes in response to IL-1β. (**A**) Phosphorylation of PRMT5 at S15 is critical for NF-ĸB activation. NF-κB luciferase assay, conducted for vector control (Ctrl), or with the overexpression of WT-PRMT5 (WT) or S15A mutant in the presence or absence of 10 ng/mL of IL-1β treatment in HEK293, and HT29, DLD1, HCT116 colon cancer cells. The data represent the means ± standard deviation (S.D.) for three independent experiments. * *p* < 0.05 vs. Ctrl+IL-1β group; ** *p* < 0.05 vs. WT + IL-1β group. (**B**) Phosphorylation of PRMT5 at S15 differentially regulates a subset of NF-κB target genes. Top panel: pie-chart (left, yellow and orange), representing data from human Illumina array assay. Data indicates that upon overexpression of WT-PRMT5, the expression of ≈48% of NF-ĸB target genes were further augmented by >= 1.5-fold following 10 ng/mL IL-1β stimulation. Among these genes, ≈39% of genes (pie-chart, right, gray and blue) could be downregulated by 2-fold or more (S15A + IL-1β/WT + IL-1β ≤ 0.5) by the S15A mutation. Bottom panel: table, showing a short list of typical NF-ĸB target genes that were upregulated by WT-PRMT5 (WT) but not by the S15A mutant. (**C**) Confirmation of Illumina Array data with qPCR analysis, indicating relative mRNA levels of CCL20 and IL8 in HEK293 and HT29, DLD1, HCT116 colon cancer cells. Ctrl: vector control cells. The data represent the means ± standard deviation (S.D.) for three independent experiments. ^†^
*p* < 0.05 vs. Ctrl group; * *p* < 0.05 vs. Ctrl+IL-1β group; ^#^
*p* < 0.05 vs. WT+IL-1β group. (**D**) Ingenuity Pathway Analysis (IPA): Subsets of genes upregulated by WT-PRMT5 overexpression but downregulated by S15A were used to conduct the IPA. Enrichment results indicating top biological functions, disease networks, and upstream regulators are shown as dots scaled by –log(p). The size of the dot shows the significant level of enrichment. (**E**) IPA representative network, showing genes regulated by S15A with NF-κB as one of the critical nodes in this network.

**Figure 3 ijms-21-03684-f003:**
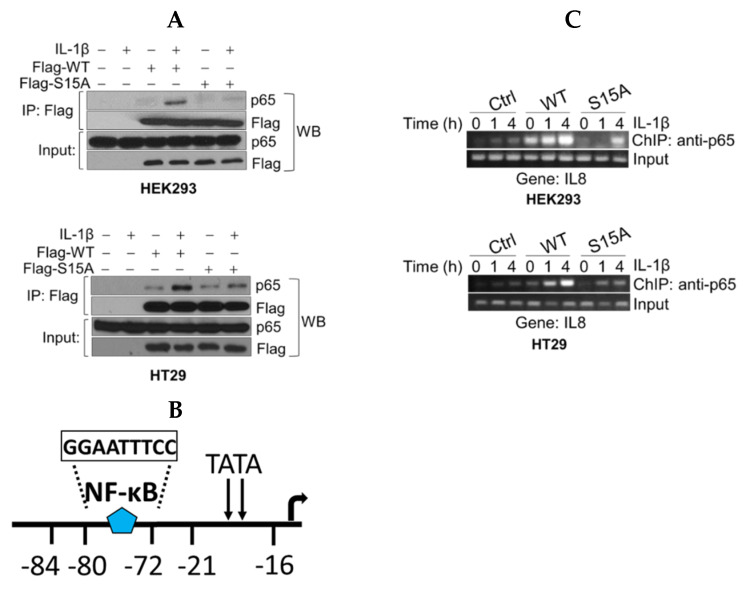
S15A mutant of PRMT5 disrupts its interaction with p65 and attenuates occupancy of p65 at NF-κB target gene. (**A**) Co-immunoprecipitation (IP) experiments, HEK293 and HT29 cells were treated or left untreated with 10 ng/mL of IL-1β for 1 h, WT-PRMT5 (Flag-WT) or S15A (Flag-S15A) was immunoprecipitated with anti-FLAG beads. Samples were then subjected to Western blot analysis (WB) and probed with anti-p65 antibody. Inputs were probed with anti-p65 and anti-Flag antibodies. (**B**) Architecture of the IL8 promoter showing the location of the NF-ĸB binding site. (**C**) Chromatin immunoprecipitation (ChIP) assay in HEK29 and HT29 cells to detect occupancy of p65 at the typical NF-κB target gene, IL8′s promoter upon IL-1β stimulation.

**Figure 4 ijms-21-03684-f004:**
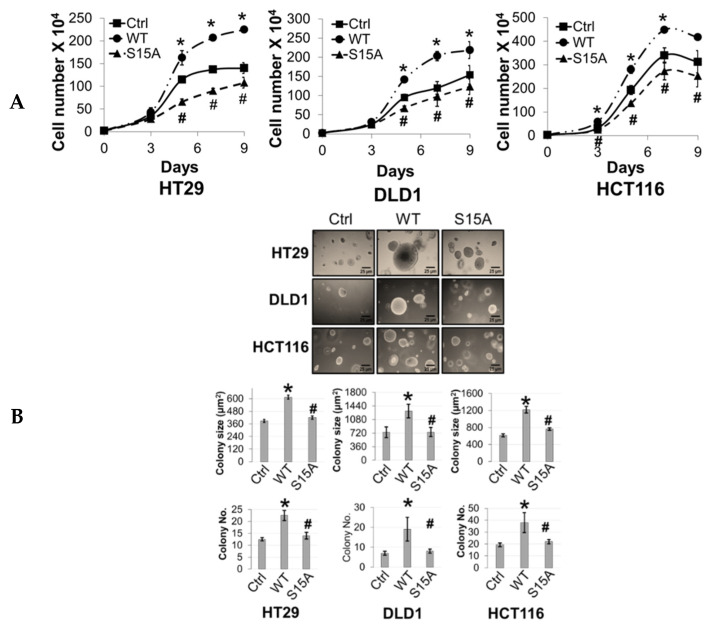
Phosphorylation of PRMT5 at S15 regulates cell growth, anchorage-independent growth and migration of colon cancer cells. (**A**) Cell growth assay compares cell numbers of vector control (Ctrl), WT-PRMT5 (WT), or S15A-PRMT5 (S15A) mutant-overexpressing cells in HT29, DLD1, and HCT116 colon cancer cells. A total of 2 × 10^4^ cells were seeded and counted using a cell counting chamber at days 3, 5, 7, and 9. The data represent the means ± standard deviation (S.D.) for three independent experiments. ** p* < 0.05 vs. Ctrl group; *# p* < 0.05 vs. WT group. (**B**) Top panel: anchorage-independent growth assay with colon cancer cells overexpressing WT or S15A mutant compared to Ctrl. Representative images of colonies for HT29, DLD1, and HCT116 are shown. Bottom panel: quantification of the average colony size and number is shown below the corresponding cell type. The data represent the means ± standard error of mean (SEM) for three independent experiments. Scale bar = 25 µm. ** p* < 0.05 vs. Ctrl group; # *p* < 0.05 vs. WT group. (**C**) Top panel: Boyden Chamber Transwell assay, showing migration of colon cancer cells overexpressing WT or S15A mutant compared to Ctrl. Representative photos of crystal violet stained cells are shown with 20× magnification. Bottom panel: quantification of the average number of migrated cells is shown. The data represent the means ± standard deviation (S.D.) for three independent experiments. ** p* < 0.05 vs. Ctrl group; ^#^
*p* < 0.05 vs. WT group.

**Figure 5 ijms-21-03684-f005:**
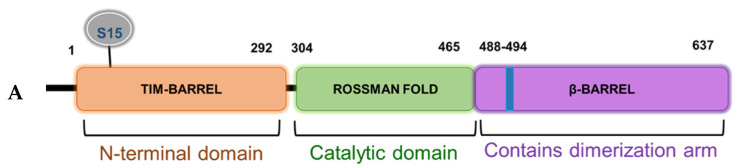
S15 phosphorylation is located in the triose phosphate isomerase (TIM) barrel domain of PRMT5 and regulates IL-1β-inducible PRMT5 methyltransferase activity. (**A**) Overview of the domain architecture of human PRMT5. The S15 residue is located in the unique N-terminal TIM-Barrel, which might be critical for PRMT5 substrate binding. (**B**) S15 phosphorylation is critical for the methyltransferase activity of PRMT5. AlphaLISA assay was conducted by using biotinylated histone H4 as a PRMT5 substrate. Graph shows detection of specific methyltransferase activity of WT-PRMT5 (WT) or S15A-PRMT5 (S15A) mutant enzymes purified from HEK293 cells in the presence or absence of 10 ng/mL of IL-1β. E444D-PRMT5 (E444D) was used as an enzymatic dead mutant control. S-adenosyl methionine (SAM) was used as the methyl donor for the reaction. The data represent the means ± standard deviation (S.D.) for three independent experiments. ** p* < 0.05 vs. WT group; ^#^
*p* < 0.05 vs. WT + IL-1β group.

**Figure 6 ijms-21-03684-f006:**
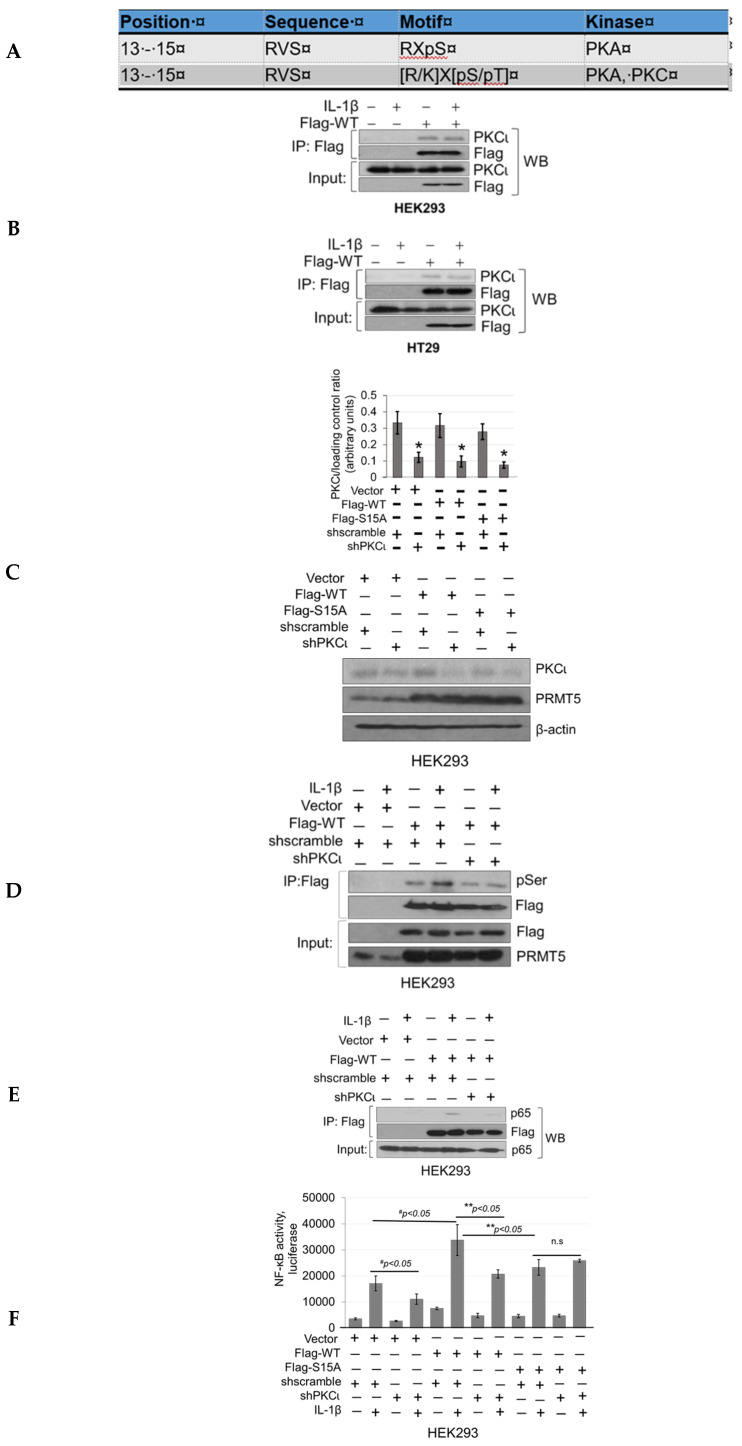
Knockdown of PKCι attenuates phosphorylation of PRMT5 and disrupts PRMT5-mediated NF-ĸB signaling. (**A**) Human Protein Reference Database (HPRD) predicted PKC and PKA phospho-motif in position 13-15 of PRMT5. (**B**) Co-immunoprecipitation (IP) experiments, HEK293 and HT29 cells were treated or left untreated with 10 ng/mL of IL-1β for 1 h, Flag-WT-PRMT5 (Flag-WT) was immunoprecipitated with anti-FLAG beads. Samples were then subjected to Western blot analysis and probed with the indicated antibodies. Inputs were probed with anti-PKCι and anti-Flag antibodies. (**C**) Establishment of vector control, WT, or S15A Flag-PRMT5 overexpressing HEK293 stable cells with or without co-expression of shscramble or shPKCι constructs. Western blot analysis, probed with anti-PKCι, PRMT5, or β-actin, respectively. Image J quantification of PKCι to loading control (β-actin) ratios for three independent Western blots is shown below. ** p* < 0.05 vs. shscramble counterparts. (**D**) Phosphorylation of PRMT5 using co-immunoprecipitation and Western blot analysis. Either HEK293 cells with vector control and shscramble, Flag-WT and shscramble, or Flag-WT and shPKCι were treated with 10 ng/mL of IL-1β or left untreated for 1 h. Samples were collected and Flag-WT was further immunoprecipitated with anti-FLAG beads and subjected to Western analysis using an anti-phospho-serine motif antibody (pSer). The inputs were probed with anti-PRMT5 or Flag antibody. (**E**) Co-immunoprecipitation (IP) experiment, HEK293 cells were treated or left untreated with 10 ng/mL of IL-1β for 1 h, WT-PRMT5 (Flag-WT) or S15A (Flag-S15A) was immunoprecipitated with anti-FLAG beads. Samples were then subjected to Western blot analysis and probed with anti-p65 antibody. Inputs were probed with anti-p65 and anti-Flag antibodies. (**F**) NF-κB activity was determined by luciferase assay, in established cells shown in Figure 6C. The data represent the mean ± SD from three independent experiments. *# p* < 0.05 vs. Ctrl+IL-1β group; ** *p* < 0.05 vs. WT+IL-1β group; n.s: not significant. (**G**) NF-κB activity luciferase assay in control or cells overexpressing WT-PRMT5 (WT) or S15A treated with or without PKCι inhibitor CRT0066854 in the presence or absence of 10 ng/mL of IL-1β stimulation in both HEK293 and HT29 cell systems. The data represent the mean ± SD from three independent experiments. # *p* < 0.05 vs. Ctrl+IL-1β group; ** *p* < 0.05 vs. WT+IL-1β group; n.s: not significant. (**H**) Top panel: box-whisker plots showing gene expression (transcript levels) of PKCι across colorectal adenocarcinoma (COAD) tumors and normal tissue based on individual cancer stages. Bottom panel: log-rank test was used to indicate statistical significance between normal and each cancer stage. Individual cancer stages are based on AJCC (American Joint Committee on Cancer) pathologic tumor stage information (http://ualcan.path.uab.edu).

**Figure 7 ijms-21-03684-f007:**
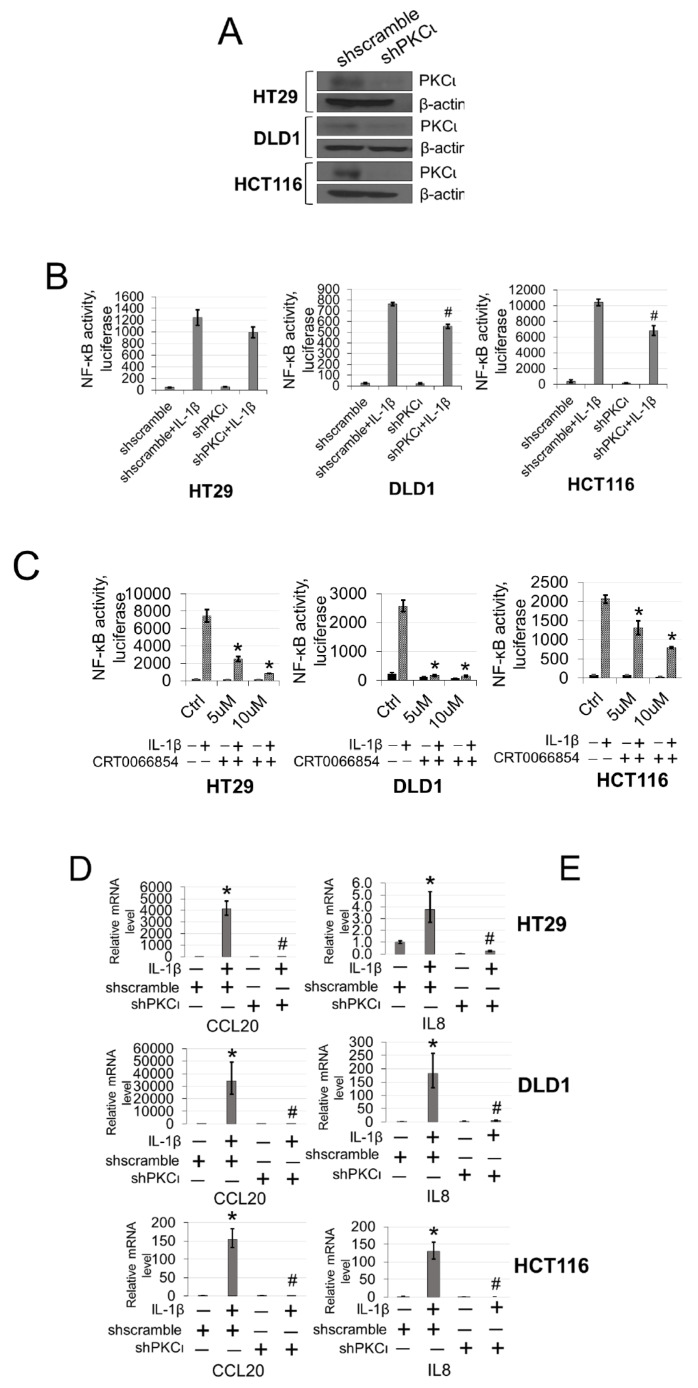
Inhibition of PKCι attenuates NF-ĸB activation and target gene expression in response to IL-1β. (**A**) Establishment of shscramble and shPKCι HT29, DLD1, HCT116 colon cancer cell lines. Western blot analysis, probed with anti-PKCι or β-actin (loading control), respectively. (**B**) Knockdown of PKCι attenuates NF-ĸB activation. NF-κB luciferase assay, conducted for shscramble and shPKCι in the presence or absence of 10 ng/mL of IL-1β in HT29, DLD1, HCT116 colon cancer cells. The data represent the means ± standard deviation (S.D.) for three independent experiments. ^#^*p* < 0.05 vs. shscramble+IL-1β group. (**C**) Small-molecule inhibition of PKCι attenuates NF-ĸB activation. NF-κB luciferase assay, conducted for vehicle control (Ctrl) or increasing µM concentrations of PKCι inhibitor CRT0066854 in the presence or absence of 10 ng/mL of IL-1β treatment in HT29, DLD1, HCT116 colon cancer cells. The data represent the means ± standard deviation (S.D.) for three independent experiments. **p* < 0.05 vs. Ctrl+IL-1β group. (**D**) qPCR analysis, indicating relative mRNA levels of CCL20 and IL8 in shscramble and shPKCι colon cancer cells in the absence and presence of IL-1β in HT29, DLD1, HCT116 colon cancer cells. The data represent the means ± standard deviation (S.D.) for three independent experiments. ** p* < 0.05 vs. shscramble group; ^#^
*p* < 0.05 vs. shscramble+IL-1β group. (**E**) qPCR analysis, indicating relative mRNA levels of CCL20 and IL8 in vehicle and PKCι inhibitor CRT0066854 treated colon cancer cells in the absence and presence of IL-1β in HT29, DLD1, HCT116 colon cancer cells. The data represent the means ± standard deviation (S.D.) for three independent experiments. ** p* < 0.05 vs. vehicle group; ^#^*p* < 0.05 vs. vehicle+IL-1β group.

**Figure 8 ijms-21-03684-f008:**
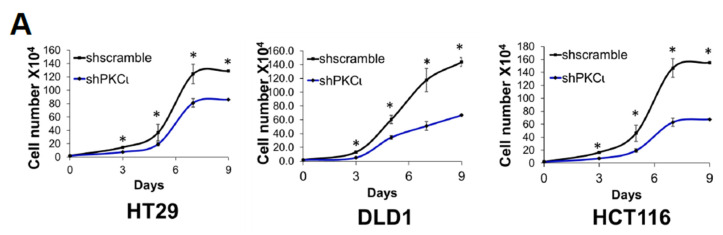
Inhibition of PKCι attenuates cell growth, anchorage-independent growth and migration of colon cancer cells. (**A**) Cell growth assay compares cell numbers of shscramble and shPKCι as well as (**B**) Vehicle control (Ctrl) and HT29, DLD1, and HCT116 colon cancer cells treated with selective PKCι inhibitor CRT0066854. A total of 2 × 10^4^ cells were seeded and counted using a cell counting chamber at days 3, 5, 7, and 9. The data represent the means ± standard deviation (S.D.) for three independent experiments. * *p* < 0.05 vs. shscramble (**A**) or Ctrl (**B**) group. (**C**) Anchorage-independent growth assay with knockdown (top panel) or small molecule inhibition of PKCι (bottom panel) compared to shscramble and vehicle control (Ctrl), respectively, in colon cancer cells. Representative images of colonies for HT29, DLD1, and HCT116 are shown. Quantification of the average colony size is shown on the right for the corresponding cell type. The data represent the means ± standard error of mean (SEM) for three independent experiments; Scale bar = 25 µm. * *p* < 0.05 vs. shscramble (top panel) or Ctrl (bottom panel) group. (**D**) Boyden Chamber Transwell assay, showing migration of shPKCι colon cancer cells compared to shscramble controls. Representative photos of crystal violet stained cells are shown with 20× magnification. Right panel: quantification of the average number of migrated cells is shown. The data represent the means ± standard deviation (S.D.) for three independent experiments. ** p* < 0.05 vs. shscramble group.

**Figure 9 ijms-21-03684-f009:**
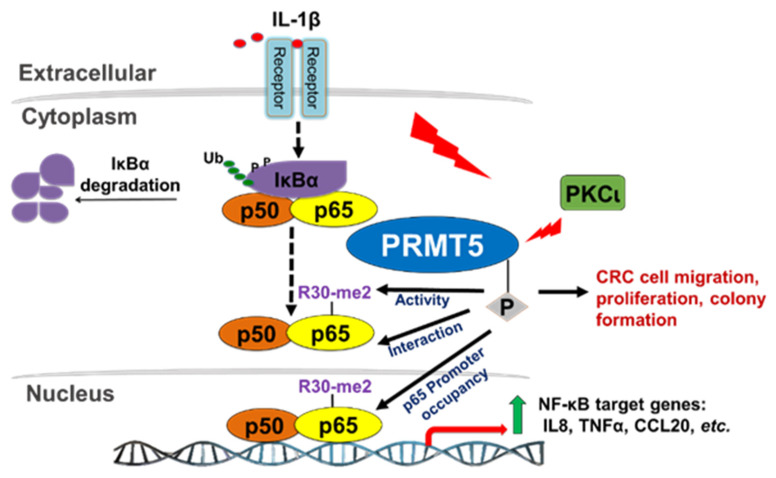
Hypothetical model. IL-1β stimulation activates the NF-κB pathway and induces PKCι-mediated phosphorylation of PRMT5. Phosphorylation of PRMT5 at S15 mediates the PRMT5-p65 interaction and proximal promoter occupancy of p65 at target genes and thus constitute pivotal mechanisms by which PRMT5 can fine-tune NF-κB activation and target gene expression. Furthermore, S15 phosphorylation mediates IL-1β-induced PRMT5 activity. Together, these potentially serve to facilitate the tumor-associated functions exerted by PRMT5-mediated NF-κB activation, including the enhanced proliferation, anchorage-independent growth, and migration associated with PRMT5 overexpression.

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
