# Peer review of "Regulation of a PRMT5/NF-κB Axis by Phosphorylation of PRMT5 at Serine 15 in Colorectal Cancer"

_ijms, 2020, doi:10.3390/ijms21103684_

Round 1
Reviewer 1 Report
In this study, Hartley et al. identified that arginine methyltransferase PRMT5 is phosphorylated at Serine 15 (Ser15) in HEK293 cells in response to stimulation with interleukin-1β (IL-1β). They then overexpressed PRMT5 wild-type (WT) or serine-to-alanine phosphorylation site mutant (S15A) and found differences in IL-1β-dependent NF-ĸB reporter activity, NF-ĸB target gene expression, PRMT5 phospho-serine levels, and PRMT5-p65 association in HEK293 cells. They also demonstrated that overexpression of PRMT5-WT but not PRMT5-S15A increased cell growth, anchorage-dependent growth, and migration of colorectal cancer (CRC) cell lines (HT29, DLD1, HCT116). Using mass spectrometry studies and a published database, the authors identified atypical PKC iota (PKCι) as the kinase phosphorylating PRMT5 at Ser15. Subsequently, they performed experiments in HEK293 and CRC cell lines (using shRNA-mediated knockdown of PKCι, or addition of PKCι/ζ inhibitor CRT0066854) showing that the IL-1β-dependent increase in NF-ĸB reporter activity was attenuated in cells overexpressing PRMT5-WT but not PRMT5-S15A.
While there is still some question as to whether the identified regulation of PRMT5 at this particular site (phosphorylation at Ser15) by this kinase (PKC iota) is truly relevant in CRC cells, the authors have adequately addressed the Reviewer’s concerns.
Minor comments:
- Information pertaining to shRNA knockdown experiments (shRNA constructs, transfections/infections, etc.) should be provided in the Experimental Procedures.
- In new Figures 7 and 8, it is evident that the CRT0066854 inhibitor has a much stronger effect on IL-1β-dependent NF-ĸB reporter activity and target gene expression, and on general cell growth of CRC cells lines, as compared to the knockdown of PKC iota (shPKCι). The authors should acknowledge this in the Discussion and provide some possible reasons for why this may be, including that CRT0066854 inhibits both PKC iota and PKC zeta isoforms.
Author Response
May 13, 2020 RE: Manuscript ID: ijms-784785 – Revision 2 (Authors: Antja Voy-Hartley et al. and Tao Lu) Dear International Journal of Molecular Sciences Editor, Thanks very much for reviewing our manuscript entitled “Regulation of a PRMT5/NF-ĸB axis by phosphorylation of PRMT5 at serine 15 in colorectal cancer”. We appreciate the reviewers’ very thoughtful suggestions. Overall, Reviewer #2 is satisfied with the Revision 1. So below we will only address the two minor issues that Reviewer #1 raised:
Minor comments:
1. Information pertaining to shRNA knockdown experiments (shRNA constructs, transfections/infections, etc.) should be provided in the Experimental Procedures. Answer: Thanks very much for the very helpful suggestion. We added this information in the Experimental Procedures, and highlighted the changes in light blue color.
2. In new Figures 7 and 8, it is evident that the CRT0066854 inhibitor has a much stronger effect on IL-1β-dependent NF-ĸB reporter activity and target gene expression, and on general cell growth of CRC cells lines, as compared to the knockdown of PKC iota (shPKCι). The authors should acknowledge this in the Discussion and provide some possible reasons for why this may be, including that CRT0066854 inhibits both PKC iota and PKC zeta isoforms. Answer: This is really very thoughtful suggestion. We added this information in the discussion section, and highlighted in light blue color.
In short, we thank the editor and reviewers for your very helpful feedback and suggestions. We hope you would agree with our effort and find this manuscript is acceptable for publication.
Thanks very much!
Sincerely, Tao Lu Tao Lu, Ph.D. (Corresponding author) 635 Barnhill Drive, MS-A521 Department of Pharmacology and Toxicology Indiana University School of Medicine Indianapolis, IN 46202, USA Tel: 317-278-0520 Fax: 317-274-7714 Email: lut@iu.edu

Reviewer 2 Report
The manuscript is well revised in the experiments and contents.
Author Response

(The authors gave the same response as above.)
